# The ESTMJS (European Society of Temporomandibular Joint Surgeons) Consensus and Evidence-Based Recommendations on Management of Condylar Dislocation

**DOI:** 10.3390/jcm10215068

**Published:** 2021-10-29

**Authors:** Andreas Neff, Niall McLeod, Frederik Spijkervet, Merle Riechmann, Ulla Vieth, Andreas Kolk, Andrew J. Sidebottom, Bernard Bonte, Bernard Speculand, Carrol Saridin, Christian T. Wilms, Constantinus Politis, David Faustino Ângelo, Dušan Hirjak, Esben Aagaard, Fabrizio Spallaccia, Florencio Monje, Gerhard Undt, Giovanni Gerbino, Hadas Lehman, Jacinto F. Sanromán, Louis G. Mercuri, Luke Cascarini, Mattias Ulmner, Maurice Mommaerts, Nadeem R. Saeed, Orhan Güven, Salvatore Sembronio, Vladimir Machoň, Linda Skroch

**Affiliations:** 1Department of Oral and Craniomaxillofacial Surgery, Philipps University Marburg, 35043 Marburg, Germany; merlee.riechmann@t-online.de (M.R.); ullaprechel@gmx.de (U.V.); l.skroch@gmx.de (L.S.); 2The European Society of Temporomandibular Joint Surgeons (ESTMJS), c/o Secretariat: 16 Hemlegh Vale, Helsby, Cheshire WA6 0DB, UK; niall.mcleod@nhs.net (N.M.); f.k.l.spijkervet@umcg.nl (F.S.); andreas.kolk@tirol-kliniken.at (A.K.); ajsidebottom@doctors.org.uk (A.J.S.); bernard.bonte@azsintjan.be (B.B.); speculand@gmail.com (B.S.); c.saridin@hagaziekenhuis.nl (C.S.); christian.wilms@mkg-troisdorf.de (C.T.W.); constantinus.politis@uzleuven.be (C.P.); david.serrano.angelo@gmail.com (D.F.Â.); hirjak.dusan@gmail.com (D.H.); eaagaard1@gmail.com (E.A.); fspallaccia@yahoo.it (F.S.); fmonje@oralmaxilofacial.com (F.M.); gerhard.undt@med.sfu.ac.at (G.U.); giovanni.gerbino@unito.it (G.G.); hlehman6@gmail.com (H.L.); fjsan@mundo-r.com (J.F.S.); lgm@tmjconcepts.com (L.G.M.); cascarini.luke@gmail.com (L.C.); mattias.ulmner@karolinska.se (M.U.); mauricemommaerts@me.com (M.M.); moodyed66@outlook.com (N.R.S.); oguven@dentistry.ankara.edu.tr (O.G.); ssembronio@hotmail.com (S.S.); machonv@seznam.cz (V.M.); 3Department of Oral & Maxillofacial Surgery, University Hospital of Leicester, Leicester LE3 9QP, UK; 4Department of Oral & Maxillofacial Surgery, University Medical Center Groningen, 9700 RB Groningen, The Netherlands; 5Department of General Medicine, UKGM GmbH, University Hospital Marburg, 35043 Marburg, Germany; 6Department of Oral and Maxillofacial Surgery, Medical University of Innsbruck, 6020 Innsbruck, Austria; 7Department of Oral and Maxillofacial Surgery, Spire Nottingham Hospital, Nottingham NG12 4GA, UK; 8Department of Oral and Maxillofacial Surgery, AZ Sint-Jan Brugge, 8000 Oostende, Belgium; 9Department of Oral and Maxillofacial Surgery, BMI Priory Hospital, Birmingham B5 7UG, UK; 10Department of Oral and Maxillofacial Surgery, Haga Ziekenhuis Hospital, University Medical Centre, 2545 AA Den Haag, The Netherlands; 11Private Practice for Oral and Maxillofacial Surgery Troisdorf, 53840 Bonn, Germany; 12Department of Oral and Maxillofacial Surgery, Leuven University Hospitals, 3000 Leuven, Belgium; 13Instituto Português da Face, 1050-227 Lisbon, Portugal; 14Faculdade de Medicina, Universidade de Lisboa, 1649-028 Lisbon, Portugal; 15Department of Oral and Maxillofacial Surgery, Comenius University, 831 04 Bratislava, Slovakia; 16Department of Oral and Maxillofacial Surgery, Kæbekirurgisk Klinik, 1120 Copenhagen, Denmark; 17Department of Maxillofacial Surgery, S. Maria Hospital Terni, 05100 Terni, Italy; 18Department of Oral and Maxillofacial Surgery, Medical School Extremadura University, 06006 Badajoz, Spain; 19Faculty of Medicine, Sigmund Freud University, 1020 Vienna, Austria; 20Department of Oral and Maxillofacial Surgery, Università degli Studi di Torino, 10124 Torino, Italy; 21Oral and Maxillofacial Surgery Unit, Shaare Zedek Medical Center, Jerusalem 9103102, Israel; 22Department of Oral and Maxillofacial Surgery, Universidad de Vigo, Povisa Hospital, 36211 Vigo, Spain; 23Department of Orthopaedic Surgery, Rush University Medical Center, Chicago, IL 60612, USA; 24Department of Oral and Maxillofacial Surgery, The Wellington Hospital, London NW8 9LE, UK; 25Department of Craniofacial Diseases, Karolinska University Hospital, 141 86 Stockholm, Sweden; 26Department of Oral and Maxillofacial Surgery, Universitair Ziekenhuis Brussel, 1090 Jette, Belgium; 27Department of Oral and Maxillofacial Surgery, Great Ormond Street Hospital for Children, London WC1N 3JH, UK; 28Department of Oral and Maxillofacial Surgery, University of Ankara, 06100 Ankara, Turkey; 29Department of Oral and Maxillofacial Surgery, Academic Hospital of Udine, 33100 Udine, Italy; 30Department of Oral Maxillofacial Surgery, Faculty Hospital Prague, Charles University, CZ-121 08 Prague, Czech Republic

**Keywords:** temporomandibular joint, joint dislocations, condylar dislocation, terminology, clinical practice guideline, evidence-based medicine

## Abstract

Although condylar dislocation is not uncommon, terminology, diagnostics, and treatment concepts vary considerably worldwide. This study aims to present a consensus recommendation based on systematically reviewed literature and approved by the European Society of TMJ Surgeons (ESTMJS). Based on the template of the evidence-based German guideline (register # 007-063) the ESTMJS members voted on 30 draft recommendations regarding terminology, diagnostics, and treatment initially via a blinded modified Delphi procedure. After unblinding, a discussion and voting followed, using a structured consensus process in 2019. An independent moderator documented and evaluated voting results and alterations from the original draft. Although the results of the preliminary voting were very heterogenous and differed significantly from the German S3 guideline (*p* < 0.0005), a strong consensus was achieved in the final voting on terminology, diagnostics, and treatment. In this voting, multiple alterations, including adding and discarding recommendations, led to 24 final recommendations on assessment and management of TMJ dislocation. To our knowledge, the ESTMJS condylar dislocation recommendations are the first both evidence and consensus-based international recommendations in the field of TMJ surgery. We recommend they form the basis for clinical practice guidelines for the management of dislocations of the mandibular condyle.

## 1. Introduction

Dislocation of the temporomandibular joint (TMJ) is not a rare event, with an estimated incidence of up to 25 of 100,000 population per year [1] and a lifetime prevalence of 5–8% [2,3,4]. Despite these diagnostics, treatment concepts vary considerably worldwide, and there is a general lack of agreement on even the basic terminology regarding dislocation of the TMJ.

Although there is evidence-based German guidelines on dislocation of the TMJ [1,5], there are none that are widely accepted internationally. Therefore, in May 2019, members of the European Society of TMJ Surgeons (ESTMJS) discussed and agreed on recommendations concerning the management of TMJ dislocation. The aim of our study is to present a state-of-the-art consensus approach to TMJ dislocation based on current literature and practical experiences of the ESTMJS members internationally.

## 2. Materials and Methods

The consensus was formed using a modified Delphi methodology [6], following the principles of the Arbeitsgemeinschaft der Wissenschaftlichen Medizinischen Fachgesellschaften (AWMF, i.e., study group of the German scientific medical societies) [7] as follows.

### 2.1. Guideline Draft

An initial set of draft guidelines were formulated, from the German S3 interdisciplinary guidelines on Condylar Dislocation (AWMF registry 007–063, June 2016 [1,5]), translated into English. This German S3 (i.e., evidence and consensus-based) guideline (first author and guideline coordinator A.N. [5]) was based on a systematic literature search, using the term “temporomandibular joint dislocation” in PubMed, Cochrane, Embase, and ZB MED databases, which was originally conducted in 2014, 2015, and 2016. The literature research methodology according to the AWMF rules for S3 guidelines [7] was basically congruent to the PRISMA checklists, and the guidelines were structured using PICOTS charts (for further details cf. long version and guideline report of the German guideline on Condylar dislocation, AWMF registry 007–063 under https://www.awmf.org/leitlinien/detail/ll/007–063.html) (accessed on 30 September 2021) [5]). Two independent authors (U.V. and L.S.) screened all papers. A third author (A.N.) was consulted in cases of disagreement between these 2 independent screening results. Papers identified were graded by their level of evidence based on the criteria of the Oxford Centre for Evidence-based Medicine [8] and according to the rules of the AWMF [7]. The literature search was repeated in 2019 (M.R., L.S., and A.N.), to assess any new publications that might affect the recommendations, and again in 2020 (M.R., L.S., and A.N.) and 2021 (M.R. and A.N.) in preparation for this manuscript.

### 2.2. Delphi Procedure (Preliminary Voting)

The initial set of draft guidelines consisted of 30 individual recommendations [5] relating to the assessment and treatment of TMJ dislocation. The expert group, consisting of members of the ESTMJS (cf. www.estmjs.org (accessed on 30 September 2021) and Appendix A), were sent this initial draft 6 weeks prior to the general assembly (GA) held in Marburg, Germany, in May 2019. They were invited to grade these and make suggestions) for alterations and modifications (i.e., blinded to the other participants), which were processed by an independent monitor (L.S.).

Grading was by means of 3 different grades of recommendation (GoR, cf. list of abbreviations in Appendix A), which take account of the level of evidence, but also expert opinion, which included clinical experience on adverse events and patient preferences (Figure 1). Accordingly, grade A stands for a strong recommendation, expressed by the word “shall”, and is usually based on studies of the highest level of evidence available (Table 1), or for which exists an extremely high consensus on good clinical practice. Grade B represents a non-emphasized recommendation based on level of evidence II (and eventually lower levels), referred to by “should.” The lowest GoR 0, phrased with “may”, leaves the decision open and is usually based on the remaining levels of evidence (LoE III-V, cf. list of abbreviations in Appendix A).

### 2.3. Final Voting (Consensus Meeting)

The results of the preliminary grading were tabulated, and an updated draft was presented at the GA. Attending members and associate members of the ESTMJS discussed and voted on this with an independent monitor (L.S.) moderating and documenting the discussion and recording results of the voting, following the rules of the structured consensus procedure of the AWMF [7].

Every voting outcome was represented by the strength of consensus (SoC, cf. list of abbreviations in Appendix A), which was based on the percentage of attendees supporting the statement (Table 2).

Statistical analysis comparing GoR, SoR, and respective alterations between German S3-Guideline, preliminary voting, and final voting was carried out, using a Wilcoxon signed-rank test. The level of significance was set at *p* < 0.05.

At the same session, the ESTMJS members also discussed and voted on definitions to be used for TMJ dislocation, aiming at establishing a uniform nomenclature of terminology.

## 3. Results

### 3.1. Literature Search and Search Update (2019–2021)

The initial literature search identified 104 relevant articles (Figure 2). Updated searches that fed into the evidence presented in the German Guidelines identified a further 34 articles. Following the new search in 2020 and February 2021, 92 newly published papers were integrated into the guideline, resulting in 230 papers that were summarized and considered in forming the present recommendation (Figure 2).

### 3.2. Participants Preliminary and Final Voting

#### 3.2.1. Participants Preliminary Voting

In the preliminary voting, 20 out of a total of 44 ESTMJS members from 12 countries returned the questionnaires; 5 members added comments to be discussed during the final voting (for more information about the ESTMJS, please cf. www.estmjs.org (accessed on 30 September 2021)).

#### 3.2.2. Participants Final Voting

In May 2019, 22 out of a potential 44 ESTMJS members from 12 countries participated in the final voting. Of these, 16 had taken part in the blinded preliminary voting, and 6 only participated in the final voting.

### 3.3. Voting Results

#### 3.3.1. Terminology

The ESTMJS members present at the GA unanimously (22/22) agreed on definitions and terminology for condylar dislocation, differentiating between (a) fixed and non-fixed dislocations, (b) single episode (one-time), recurrent and habitual dislocations and (c) acute, chronic (persistent) and longstanding dislocations (Table 3).

#### 3.3.2. Initial Draft

There were 30 recommendations made, derived from the German guidelines on the management of condylar dislocation. There was a strong consensus on all of these reached by the approving committee.

#### 3.3.3. Preliminary Grading

Of the 30 original recommendations there was disagreement with only 3 (9%), although no strong consensus (approval >95% of all participants) could be reached on the remainder (Appendix A). The difference in SoC between the German guideline and the Preliminary Voting carried out by ESTMJS-members was statistically significant (*p* = 0.0005) (Figure 3). There were 30 recommendations carried forward for discussion and voting at the GA.

#### 3.3.4. Final Voting

Of the 30 recommended proposals, 8 draft recommendations were discarded after discussion, whereas 2 were newly added, leading to 24 consensus-based recommendations (Table 4). Twelve recommendations were accepted with strong consensus, without any alterations. For 6 recommendations, a strong consensus was reached after a modification of the text, and for 4 recommendations, the members reached a strong consensus for a different grade of recommendation (GoR).

At this point, all 24 recommendations were approved with strong consensus. There was a statistically significant difference regarding GoR and content (changes of text, deleted and new passages) between the original German guideline and the final version approved by ESTMJS in its modified form at the GA (*p* = 0.0005, Wilcoxon signed-rank test), despite both reaching strong consensus (with SoC = 100%) regarding all, respectively, adopted recommendations.

Concerning Grades of recommendation, the ESTMJS members in most cases (15/24) agreed on a recommendation GoR B (“should”), whereas in 8/24 recommendations, there was an agreement on GoR 0, i.e., an open recommendation (see above). Overall, only one strong recommendation displayed by GoR A (“shall”) was established.

Regarding SoCs, there was a significant difference between the German guideline (all recommendations made with strong consensus) and the preliminary voting (*p* < 0.005). In contrast to the preliminary voting, for which none of the recommendations achieved a strong consensus, the final voting established a strong consensus for all recommendations (i.e., matching SoC of the German guideline) after a moderated discussion and modification of various topics during the GA. This included both changes of text and GoR.

We evaluated the changes between SoCs and GoR in the preliminary voting and the approved final version of recommendations.

Recommendations with a consensus in the initial voting were more likely to be left unchanged (8/14, 57%). Of the reminder with consensus initially, 2/14 (14%) reached strong consensus with some modification and 4/14 (29%) were ultimately discarded.

In recommendations “approved by majority,” 6/13 (46%) were unchanged in the final recommendations, 5/13 (38%) were modified (2 had wording changes and 3 changes to GoR), and 2/13 (15%) were discarded.

Of the 3 recommendations for which no consensus was reached initially, 2 were discarded and 1 achieved strong consensus with a change in wording and GoR.

## 4. Discussion

Dislocation of the mandibular condyle is not a rare occurrence, despite which there is a conspicuous lack of sound epidemiological data available. It has been reported to account for 3% of all articular dislocations of the body [9] and can occur in every age group [2]. Dislocation may be missed where there are co-morbidities or communication difficulties, and it can be assumed that many cases are unreported [1]. Trauma is described as the primary cause in anything from 6% to 60% of cases, with spontaneous dislocation and dislocation secondary to intubation (anesthetic or endoscopy) being the other most common causes [4,10,11]. Certain morphologic features of the temporomandibular joint, such as a flat mandibular condyle, may increase the risk of dislocation even with daily activities, such as yawning [3,12]. Laxity of the joint capsule or ligaments may predispose to dislocation, particularly in the elderly [11,13]. Neurologic disorders, particularly in elderly patients, may give rise to chronic dislocation and need particular attention [14]. A shortened dental arch may also predispose to dislocation [15,16].

Early, correct diagnosis is important to allow for immediate treatment, with the highest likelihood of success [4,17]. In longstanding (i.e., persisting) dislocations, manual reduction is seldomly successful, and surgical treatment may be required [3,18]. Moreover, patients with delayed treatment are more often prone to recurrent dislocations [19].

To the best of our knowledge, there are no international guidelines or consensus-based recommendations regarding the nomenclature and management of dislocation of the temporomandibular joint, despite the plethora of literature on the topic [1,3,10,12,20,21]. There is, therefore, a strong need for an evidence-based consensus on the nomenclature and management of dislocation of the temporomandibular joint.

With regards to nomenclature, dislocation of the temporomandibular joint is inconsistently described in the literature, and many different terms have been used interchangeably. The ESTMJS members therefore discussed and ultimately reached a strong consensus on an appropriate nomenclature to be used.

First of all, what is meant by a dislocation must be clear. In most studies, the term is used for a fixed displacement of the mandibular condyle out of the articular fossa, which must be reduced by a trained physician [1,10]. These episodes can be differentiated from a non-fixed displacement of the mandibular condyle, which reduces spontaneously or can be self-reduced, which are best termed “subluxations” [1]. The meaning of one-time dislocation is self-evident, with the term recurrent applying when there has been any more than a single episode. The term “habitual dislocation” is furthermore defined as where dislocation occurs during physiological movements such as chewing, speaking, etc., which typically occurs after a series of recurrent dislocations. Most authors classify TMJ dislocations into acute, chronic, and recurrent, however, there is no consistent definition for those terms. In most studies, the term “acute” is used for (single episode) dislocations [1,10,11]. There is much less consensus on what defines “chronic dislocation”. While Hillam and Isom [22] labeled a dislocation as “chronic” after 72 h, Papoutsis et al. [11] described occurrences persisting for more than 72 h as “chronic persistent” and as “longstanding” or “protracted” cases persisting more than a month. For Akinbami [10], a “chronic protracted” dislocation commenced after two weeks, when spasms and shortening of the temporalis and masseter muscles occur, and reduction becomes difficult to achieve manually. “Acute” is best considered a recently occurred dislocation, with “chronic” defined as where this has persisted for more than 4 weeks. Degenerative changes to the condyle and surrounding soft tissue may be seen in chronic dislocation, and to highlight where such changes are found, the term “longstanding” is proposed, defined primarily by the pathological changes rather than duration per se [23].

With regards to clinical assessment of temporomandibular joint dislocation, in the absence of acute facial trauma, the diagnosis of dislocation may be made solely on medical history and physical examination (inspection, palpation) [17,24]. Radiographs are not required in standard cases but should be considered in patients with atypical symptoms or a history of facial trauma [25]. These may also be indicated in the post-acute phase for assessing pathogenesis and for considering further therapeutic approaches [10].

After the diagnosis is established, manual reduction of the dislocation may be initially attempted without administration of any medications [10,12,26]. If such attempts are unsuccessful, further attempts should be made with the aid of muscle relaxants and/or analgesics, and if required, with the use of sedation or general anesthesia [10,12,26,27]. This attempt at manual reduction should initially be made according to the Hippocratic method of reduction, as it has been demonstrated to have a high rate of success [10,28]. Although the wrist pivot method of reduction is described with a comparatively high level of evidence (LoE Ib-IV) to be at least equal to the Hippocratic method of reduction [17,25], the ESTMJS members were not experienced with the technique, and only 2/20 were prepared to support this alternative recommendation in the draft guidelines. Furthermore, when using the Hippocratic method, it was strongly recommended that the physician’s thumbs should be placed on the oblique line of the mandible instead of the patient’s molars or using bite blocks and/or gloves to prevent biting injury during manual reduction [26,29].

In patients with potentially infectious diseases, dementia, etc., unilateral dislocation reduction may also be performed via the extraoral route [25].

A topic that generated much controversy was the sequencing of the repositioning technique. Whereas the German S3 Guideline strongly recommends (GoR A, SoC: strong consensus, i.e., “must”) performing manual reduction one side at a time, as suggested by some authors [27,30], the ESTMJS members felt strongly that the decision to reduce either unilaterally first or synchronously bilaterally should be left to the individual preference and experience of the physician (GoR 0, SoC: strong consensus).

Another controversial topic identified during the consensus session was the optimum position of the patient during the repositioning maneuver. Recent publications recommend a manual reduction of acute non-traumatic temporomandibular joint dislocations in a supine position [31,32]. In contrast, the German S3 Guidelines recommended that a manual reduction is attempted in a sitting position with the patient’s head stabilized on a headrest, as described by Chan et al. [26] and Chen et al. [33]. Some ESTMJS members stated a personal preference to stabilize the patient’s head against their sternum instead of the headrest, emphasizing the need for stabilization, and ultimately it was considered that the technical details of the respective repositioning techniques are of lesser relevance than their being performed competently and efficiently and that the repositioning of acute non-traumatic dislocations can be performed in several ways, dependent on the physician’s clinical expertise and experience (GoR 0, SoC: strong consensus). Bandages may be used post-reduction to aid stabilization but should be considered in cases of recurrent, longstanding, and/or habitual dislocations.

Non-surgical methods should have failed before any minimally invasive or open surgical intervention is considered in acute dislocation [4,10]. Minimally invasive techniques for recurrent dislocation include Botulinum toxin injection [34,35,36,37], sclerotherapy [24,38,39], and autologous blood injection (ABI) [40,41,42,43,44,45,46,47]. The best evidence thus far is for the use of ABI with level of evidence Ib [40]. Immobilization may be indicated after autologous blood injection therapy [40,41,48] with the aim of limiting the maximum opening of the jaws; rigid fixation is not recommended.

Surgical techniques should be considered where non-surgical and minimally invasive techniques are not successful in avoiding recurrent dislocation [14]. Surgical methods for treatment of recurrent dislocations include eminectomy to facilitate spontaneous reduction [48,49,50,51,52,53], restrictive techniques for prevention of recurrence of dislocation (blocking or redressment procedures) [54,55,56,57], and surgical correction of capsular ligament complex [58,59,60,61]. After any surgical treatment, patients should, for a few days, eat soft foods only and refrain from opening the mouth widely [17,24,41,50,57,60]. After surgery on the capsular ligament complex [48,59,60] immobilization may also be indicated.

In the preliminary voting, the highest level of agreement (19/20) was on the statement that patients with persisting dislocation should be treated according to an individualized protocol considering the entire range of available surgical methods and procedures.

Another controversy focused on condylar dislocation potentially occurring during intubation for general anesthesia. The most frequently described iatrogenic cause for temporomandibular dislocation is general anesthesia with oral intubation as well as endoscopic and laryngoscopic procedures requiring wide mouth opening [3,62,63]. In the German S3 Guideline, there was a unanimous recommendation to evaluate the individual patient’s risk for dislocation prior to any intubation as well as clinically check functional jaw mobility before and after such measures [62,63]. The ESTMJS members, however, discarded these recommendations, on the basis that they were more for consideration by anesthetists as part of pre-anesthetic assessment.

The blinded preliminary vote methodology of the modified DELPHI-procedure allows for anonymous comments and remarks and heightens the individual’s opinion, and, therefore, serves to collect all themes worth discussing and identifying initial trends. It can, however, result in significant heterogeneity in voting, as demonstrated here where there was significant discordance within the preliminary voting, especially when compared to the homogeneity achieved with the German S3-guideline, which served as a draft document. This may in part be due to the therapeutically more homogenous German guideline task force, which consisted of 9 members instead of the 22 ESTMJS members. At the general assembly of the ESTMJS, participants were conversely exposed to interpersonal social, and situational factors of influence, including the non-blinded immediate result of the poll, resulting in potential bias by peer group pressure. The moderated consensus session offered the opportunity to discuss, reassess, and change various recommendations or text passages, leading to a draft that could be agreed on unanimously. An independent moderator was used in our study according to the rules of the AWMF [7] to prevent over-emphasis of certain opinions in the discussion. Although there were ultimately significant differences between the draft proposals and the agreed final recommendations of this guideline, there was complete consensus from members of the ESTMJS on this final version, based on the most recent literature and the experience of this expert group. A full version of the ESTMJS guideline (including guideline report) can be found on www.estmjs.org (accessed on 30 September 2021).

Whilst they may not have legally binding status, we strongly recommend these guidelines for clinical use by all clinicians involved in the management of TMJ dislocation. Furthermore, the ESTMJS condylar dislocation project hereby also has proven to work as a pilot for establishing further evidence and consensus-based international recommendations, and thus may help to establish evidence-based diagnostics and therapy for patients according to a pan-European and/or international consensus.

Nevertheless, there are several limitations to this study. While the German S3 guideline may well be considered representative on a national level, the ESTMJS recommendations are not fully equivalent on an international level. In contrast to the German guidelines, which are issued by the AWMF as an umbrella organization of the national scientific societies and are formally approved by the respective national boards of the interdisciplinary societies involved, the ESTMJS rather constitutes a more or less random sample of individuals. In addition, not all ESMTJS members participated actively in establishing the consensus. Even though the ESTMJS members are experts in the field of TMJ, mostly based at universities and/or scientifically active, they do not represent all European TMJ surgeons and do not hold a formal mandate to represent their national OMFS societies. As a consequence, the results of our study are less representative and robust and may not be generally applicable for other countries worldwide. They may, however, well constitute evidence of a trend. In a next step, therefore, this guideline should strive for approval by the respective national European societies.

## 5. Conclusions

To our knowledge, the condylar dislocation recommendations are the first both evidence and consensus-based international management recommendations in the field of TMJ surgery established according to a well-defined and proven consensus protocol. The recommendations were accepted by the ESTMJS, representing a scientific society of European and international experts in the field of TMJ surgery. The ESTMJS condylar dislocation recommendations may thus be representative for state-of-the-art handling and managing temporomandibular joint dislocations according to current European standards.

The codified consensus approach according to the German guidelines protocol can also be recommended for further guidelines due to its evidence-based, transparent, and properly defined procedure.

## Figures and Tables

**Figure 1 jcm-10-05068-f001:**
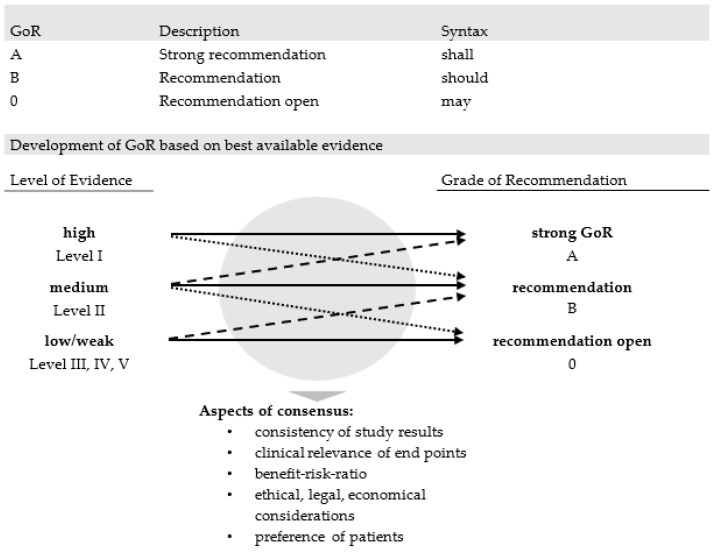
Grades of recommendation depending on best available evidence.

**Figure 2 jcm-10-05068-f002:**
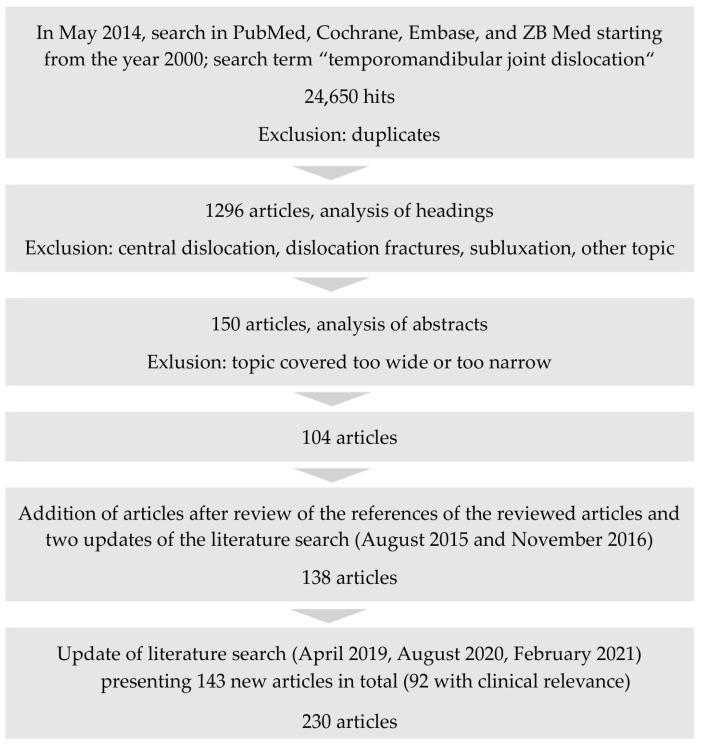
Literature search.

**Figure 3 jcm-10-05068-f003:**
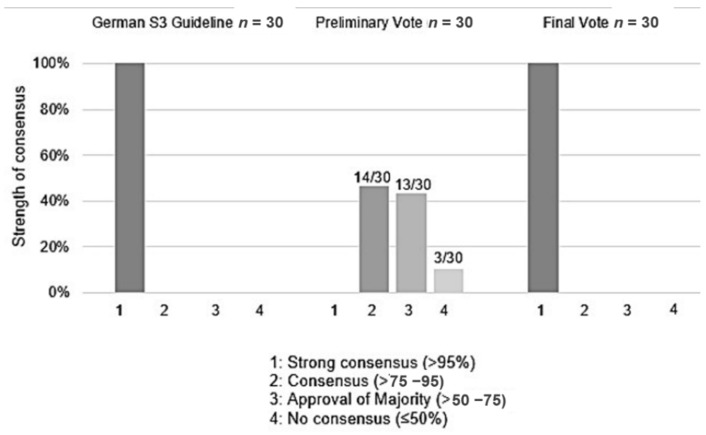
Comparison of percentage of Strength of Consensus (SoC) in the German S3 Guide-lines, preliminary as well as final vote.

**Table 1 jcm-10-05068-t001:** Criteria for Evidence classification (Oxford Center of Evidence-based medicine).

Grade of Evidence		Study Design
I	a	Meta-analysis/systematic review of GoE I papers
	b	Randomized controlled clinical trial (RCT)
II	a	Meta-analysis/systematic review of GoE II papers
	b	Controlled clinical trial (CCT)/experimental study with the control group (prospective)
III	a	Meta-analysis/systematic review of GoE III papers
	b	Retrospective cohort study, retrospective case-control study
IV	a	Meta-analysis/systematic review of GoE IV papers
	b	Non-controlled case series (<1 subject), animal experiment
V	a	Meta-analysis/systematic review of GoE V papers
	b	Case report, expert opinion
	+	Good quality or sample size *n* > 100
	−	Poor quality or sample size *n* < 10

**Table 2 jcm-10-05068-t002:** Strength of Consensus.

strong consensus	>95% of participants
consensus	>75–95% of participants
approval by the majority	>50–75% of participants
no consensus	≤50% of participants

**Table 3 jcm-10-05068-t003:** Nomenclature of TMJ dislocation approved by general assembly of ESTMJS.

1. Self-reducibility		
	fixed	not self-reducible, needs manual reduction (medical intervention)
	non-fixed/(“subluxation”)	spontaneously self-reducible
2. Occurrence of dislocation over time		
	one-time	single episode
	recurrent	multiple dislocations over time
	habitual	dislocations during physiological movements
3. Duration of dislocation		
	acute	recently occurred dislocation
	chronic/persistent	dislocation more than 4 weeks
	longstanding	adaptive/degenerative changes in or around the joint

**Table 4 jcm-10-05068-t004:** Final recommendation in full.

Examinations	
1	Patients without acute facial trauma who for the first time experience a temporomandibular joint dislocation may be diagnosed based on medical history and physical examination (inspection, palpation) if the symptoms are sufficiently indicative of a temporomandibular joint dislocation.
2	X-rays are not mandatory in standard cases, but imaging examinations should be considered in patients with symptoms allowing for other differential diagnoses to rule out facial fractures and to provide information for further treatment planning.
3	These optional additional alternative examination methods may be indicated in the post-acute phase for the purposes of revealing pathogenesis and for appraisal of further therapeutic approaches.
Treatment	
4	An attempt at a manual reduction should initially be made in cases of non-traumatic temporomandibular joint dislocation. The earlier reduction is performed, the greater the chances for a successful reduction.
5	The ESTMJS members have no experience with these alternative repositioning procedures described in the literature. The ESTMJS members, therefore, recommend that any attempt at a manual reduction should initially be made according to the Hippocratic method of reduction, as it has demonstrated a high rate of success according to literature.
6	Reduction may be performed separately one side at a time or bilaterally.
7	In the literature, there is a recommendation to use of bite blocks and double gloves to help to prevent bite injuries and associated infections. ESTMJS Members recommend the thumbs should be put on the oblique line instead.
8	If a reduction is to be performed with the patient in a sitting position, the patient’s head should be stabilized.
9	The attempt at a manual reduction of an acute dislocation may initially be made without the administration of any medications.
10	If such attempts are unsuccessful, further attempts should be made under medication (muscle relaxants and/or analgesics) and, if required, under analog sedation or under general anesthesia.
11	In patients with potential infectious diseases, dementia, etc., unilateral dislocation reduction may also be performed via the extraoral route
12	In acute dislocations, bandages may be used after reduction to help maintain stabilization.
13	In cases of recurrent, longstanding and/or habitual dislocations, securing methods should be considered.
14	Non-surgical methods should have failed before any minimally invasive or open-surgical intervention.
15	Treatment of recurrent/persistent temporomandibular joint dislocation with botulinum toxin thus far remains an off-label use of the drug. Warnings of the manufacturers regarding the use of and indications for botulinum toxin shall be observed.
16	The authors of this recommendation, however, are of the opinion that the use of botulinum toxin for the treatment of recurrent dislocations should be included as a potential indication.
17	If reduction by non-surgical methods remains unsuccessful, e.g., in cases of longstanding dislocations, surgical methods should be considered.
18	Also, in patients with *recurrent* dislocations, an indication for open surgical treatment should be established after the failure of non-surgical treatments and/or minimally invasive therapy.
19	The small sample size, varying follow-up periods, and inhomogeneous target parameters render comparisons and evaluation of long-term effects difficult (damage, recurrence of dislocation). Especially in cases of persisting dislocations, an individualized approach based on the entire range of available surgical methods and procedures should be considered.
20	After any surgical treatment, patients should, for a few days, eat soft foods only and refrain from opening mouth widely.
21	Moreover, especially after autologous blood injection therapy and after surgery on the capsular ligament complex immobilization may be indicated. The goal is here to limit the maximum opening of the jaws; rigid fixation is not recommended.
22	In case of secondary damage such as malocclusion (e.g., anterior open bite due to persisting dislocations refractory to treatment), an individualized approach based on the range of functional surgical procedures for temporomandibular joints as well as reconstructive and orthognathic surgery may be required.
Recommendations	
23	The treatment of temporomandibular joint dislocation should be initiated as early as possible to limit degenerative changes or their progression, resulting from recurrent dislocation or increasing dislocation rate, and thus to enhance the chances of success of conservative/minimally invasive treatment methods.
24	The treatment which has the best chance of success is dependent on numerous factors (pathogenesis, age of patient, secondary diagnoses, compliance, treatment goals, care structures, among others). Thus, the best treatment for each individual patient should be determined based on a thorough medical history and physical examination.

## Data Availability

The data presented in this study are available on request from the corresponding author.

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
