# Peer review of "The ESTMJS (European Society of Temporomandibular Joint Surgeons) Consensus and Evidence-Based Recommendations on Management of Condylar Dislocation"

_jcm, 2021, doi:10.3390/jcm10215068_

Round 1

Reviewer 1 Report

Congratulations for the work done.

This statement: "Although there are evidence based German guidelines on dislocation of the TMJ, there are none that are widely accepted internationally", It implies that there are no other reference guides or publications on the management of this pathology... It is worth highlighting works such as:

  • Diagnostic Criteria for Temporomandibular Disorders (DC / TMD) for Clinical and Research Applications: recommendations of the International RDC / TMD Consortium Network * and Orofacial Pain Special Interest Group. Schiffman E et al. J Oral Facial Pain Headache. 2014 Winter; 28 (1): 6-27. doi: 10.11607 / jop.1151.
  • Surgical management of recurrent TMJ dislocation-a systematic review. Tocaciu S, McCullough MJ, Dimitroulis G. Oral Maxillofac Surg. 2019 Mar; 23 (1): 35-45. doi: 10.1007 / s10006-019-00746-5. Epub 2019 Feb 7.
  • [Experts consensus on MRI examination specification and diagnostic criteria of temporomandibular joint disc displacement]. Fu KY, Hu M, Yu Q, Yang C, Cheng Y, Long X, Zhang ZG, Liu HC. Zhonghua Kou Qiang Yi Xue Za Zhi. 2020 Sep 9; 55 (9): 608-612. doi: 10.3760 / cma.j.cn112144-20200514-00268.

I would like to know what is the reason for the delay between the review of the literature, the consensus of recommendations and the performance of the Delphi method.

The literature search approach should be presented with further explanation. Was any specific methodology followed (PICOTS, PRISMA)? What criteria were followed for the choice of papers?

who made the "initial set of draft guidelines"?

the descriptive about the voting participants could be included? senior doctors? proffesors? which countries were included?

image quality should improve. I don't see table 1 necessary. Table 4 could be included as supplementary material

I miss a section of limitations. What was the reason for the low participation in the Method? Perhaps the means of communication between participants was not the ideal one? How much time was left to answer?

Author Response

First, we would like to thank you for taking time to review our paper. Many thanks also for providing suggestions for three papers to be considered with regard to existing international recommendations on condylar dislocation.

Query: This statement: "Although there are evidence based German guidelines on dislocation of the TMJ, there are none that are widely accepted internationally", It implies that there are no other reference guides or publications on the management of this pathology... It is worth highlighting works such as:

  • Diagnostic Criteria for Temporomandibular Disorders (DC / TMD) for Clinical and Research Applications: recommendations of the International RDC / TMD Consortium Network * and Orofacial Pain Special Interest Group. Schiffman E et al. J Oral Facial Pain Headache. 2014 Winter; 28 (1): 6-27. doi: 10.11607 / jop.1151.
  • Surgical management of recurrent TMJ dislocation-a systematic review. Tocaciu S, McCullough MJ, Dimitroulis G. Oral Maxillofac Surg. 2019 Mar; 23 (1): 35-45. doi: 10.1007 / s10006-019-00746-5. Epub 2019 Feb 7.
  • [Experts consensus on MRI examination specification and diagnostic criteria of temporomandibular joint disc displacement]. Fu KY, Hu M, Yu Q, Yang C, Cheng Y, Long X, Zhang ZG, Liu HC. Zhonghua Kou Qiang Yi Xue Za Zhi. 2020 Sep 9; 55 (9): 608-612. doi: 10.3760 / cma.j.cn112144-20200514-00268.

Answer: We gladly integrated the second mentioned review article (Tocaciu et al., 2019) into our article (which is also part of our updated ESTMJS version), whereas we found the other two papers not really matching our topic. The first paper (Diagnostic Criteria for Temporomandibular Disorders (DC / TMD by Schiffmann et al.) exclusively covers the diagnostic criteria of “subluxation” (viz. condylar hypermobility, however, not condylar dislocation) and does not give treatment recommendations on condylar dislocation. The third work mentioned (Fu et al., 2020) - at least according to the abstract and title – appears to deal with dislocation of the condylar disc, if we are not mistaken (unluckily we are not able to read Chinese papers).

We added (lines 268-271, in red): To the best of our knowledge there are no international guidelines or consensus-based recommendations regarding the nomenclature and management of dislocation of the temporomandibular joint, despite the plethora of literature on the topic [1, 3, 10, 12, 20, 21].

Furthermore: To our knowledge, the condylar dislocation recommendations are the first both evidence and consensus based international management recommendations in the field of TMJ surgery established according to a well-defined and proven consensus protocol.

Query: I would like to know what is the reason for the delay between the review of the literature, the consensus of recommendations and the performance of the Delphi method.

Answer: The original draft of recommendations (as a base for discussion) was based on the translated German guideline from 2016 (with respective systematic literature review following AWMF and Oxford guidelines). In 2019, in preparation of the Delphi procedure among the ESTMJS members, a systematic literature update was carried out (LS, MR and AN) and all updates retrieved for the timespan between 2016 and 2019 were made transparent to the ESTMJS members and discussed by the participants of the ESTMJS during the consensus meeting and accordingly changes were performed with regard to the German guideline recommendations from 2016. Thus, in 2019, all recommendations were up to date to the then existing literature. Next, data were processed and underwent a statistical analysis. As our manuscript underwent several revisions by the co-authors including major language editing by second author NML as native speaker, publication of our results unluckily was delayed. Therefore, to keep our literature up to date, another literature search was performed in 2020 (LS, MR and AN) and a final update in 2021 (MR and AN) before finalizing the paper. Before submitting to JCM all recommendations established in 2019 were once more checked for plausibility and approved by all of our co-authors before submission.

We exchanged Figure 2, updating the 2021 literature update results and modified our methods part to make (a) the timepoints of literature updates more transparent and to explain better the different stages of our drafts (please cf. next query)

Query: The literature search approach should be presented with further explanation. Was any specific methodology followed (PICOTS, PRISMA)? What criteria were followed for the choice of papers?

Answer: Our literature search was carried out in accordance to the rules of the Arbeitsgemeinschaft der Wissenschaftlichen Medizinischen Fachgesellschaften (AWMF, i.e. study group of the German scientific medical societies). These rules for literature search for evidence-based guidelines (S3) are basically congruent to the PRISMA guidelines for systematic reviews and are based on either the Oxford criteria (as we used) for assessment or SIGN (we used Oxford, which is better fitting for our topic with a majority of lower levels of evidence, which we also included into our search results). For the German guideline, PICOTS charts (required already for registration by the AWMF) were formulated and are presented in the respective version published by the AWMF (cf. source 5, which was the basis for our draft version). Instead of registering e.g., under PROSPERO, the 2016 German guideline activities were registered and published nationally (homepage AWMF) by the AWMF (at present, an update of the German guideline is currently running the interdisciplinary consensus procedure, as all S3 German guidelines need to be re-evaluated after 5 years as a latest and another literature update thus was performed in 2021 congruent to our paper, cf. https://www.awmf.org/leitlinien/detail/ll/007-063.html,you may also find a detailed report on the methodology of the German guideline, there.)
With some amendments in the text, we hope that our method of literature search will now be more transparent and understandable.

We modified :

  1. Materials and Methods

The consensus was formed using a modified Delphi methodology [6], following the principles of the Arbeitsgemeinschaft der Wissenschaftlichen Medizinischen Fachgesellschaften (AWMF, i.e. study group of the German scientific medical societies) [7] as follows;

2.1. Guideline draft

An initial set of draft guidelines were formulated, from the German S3 interdisciplinary guidelines on Condylar Dislocation (AWMF registry 007-063, June 2016 [1;5]), translated into English. This German S3 (i.e., evidence and consensus-based) guideline (first author and guideline coordinator A.N. [5]) is based on a systematic literature search, using the term “temporomandibular joint dislocation” in PubMed, Cochrane, Embase, and ZB MED databases which was originally conducted in 2014, 2015 and 2016. The literature research methodology according to the AWMF rules for S3 guidelines [7] is basically congruent to the PRISMA checklists and the guidelines are structured using PICOTS charts (for further details cf. long version and guideline report of the German guideline on Condylar dislocation, AWMF registry 007-063 under https://www.awmf.org/leitlinien/detail/ll/007-063.html) [5]). Two independent authors (U.V. and L.S.) screened all papers. A third author (A.N.) was consulted in cases of disagreement between these two independent screening results. Papers identified were graded by their level of evidence based on the criteria of the Oxford Centre for Evidence-based Medicine [8] and according to the rules of the AWMF [7]. The literature search was repeated in 2019 (M.R., L.S. and A.N.), to assess any new publications that might affect the recommendations, and again in 2020 (M.R., L.S. and A.N.) and 2021 (M.R. and A.N.) in preparation of this manuscript.

2.2. Delphi procedure (preliminary voting)

The initial set of draft guidelines consisted of 30 individual recommendations [5] relating to the assessment and treatment of TMJ dislocation. The expert group, consisting of members of the ESTMJS (cf. www.estmjs.org and eTable 1), were sent this initial draft 6 weeks prior to the general assembly (GA) held in Marburg, Germany, in May 2019. They were invited to grade these and make suggestions) for alterations and modifications (i.e., blinded to the other participants), which were processed by an independent monitor (L.S.).

We then also added a subsection header for the 2.3 final voting (consensus meeting), cf. query by reviewer#2

Query: who made the "initial set of draft guidelines"?

Answer: The initial draft of the recommendations to work as a base for discussion during the Delphi procedure was created by A.N., who is also first author of the German guidelines 2016. We added this information to our manuscript (please see above).

In addition we would like to note that the recommendations of the German S3 guideline on Condylar dislocation are the results of an interdisciplinary task force with members holding an official mandate of the cooperating scientific societies (cf. https://www.awmf.org/leitlinien/detail/ll/007-063.html (cf. source (5))

We also mentioned this difference to the AWMF procedure in the limitations section (interdisciplinary approach, task force members holding official mandates of the involved scientific societies), cf. queries of reviewer #2

Query: the descriptive about the voting participants could be included? senior doctors? proffesors? which countries were included?

Answer: For the description of the ESTMJS-members, we have added a link to the homepage of the ESTMJS, where all data regarding affiliations and  current functions and positions are given in detail. For basic information, please see the affiliations in the list of authors.

Please note that ESTMJS members are recognized experts in the field of TMJ surgery and become members by invitation, only.  The recommendations, therefore, may well be considered as recommendations given by an international group of European experts in the field of TMJ surgery. Furthermore, a supplementary table with all the participating members and respective country of origin is available as supplementary material in the acknowledgements.

Query: image quality should improve. I don't see table 1 necessary. Table 4 could be included as supplementary material

Answer: One of the suggestions was to discard table 1; we politely ask the reviewer to reevaluate this recommendation, because we hold that the manuscript and reader should benefit from Table 1, as it helps to easily explain and visualize the Oxford criteria, but we agree about Table 4 and will include it as supplementary material (as eTable 3). If required, we could also move Table 1 to the supplementary files, but would plead for keeping it in the running text.

Query: I miss a section of limitations. What was the reason for the low participation in the Method? Perhaps the means of communication between participants was not the ideal one? How much time was left to answer?

Answer: This criticism most probably may allude to the fact that not all ESTMJS members participated in the Delphi and consensus procedure. Please keep in mind that  the consensus meeting performed in 2019 (pre Corona) required a personal presence at the annual meeting (the consensus meeting in fact took a whole day of vivid discussions) and that it is rather unusual that all members of an international society will be able to attend an General Assembly (GA) in person (those who attend usually are the more active society members, who were also willing to participate during the Delphi procedure). Six weeks prior to the annual meeting (and therefore, the final voting), the original draft was sent out to all ESTMJS-members as a basis for discussion and the ESTMJS members were expected to prepare themselves for the consensus meeting based on the draft (to be handed in before the consensus meeting as a latest), accordingly and in doing so also  to participate in the Delphi procedure. Moreover, only those members who participated in the GA  could participate in the final voting.

We added a section: “Limitations of the study”:

 Nevertheless, there are several limitations to this study. While the German S3 guideline may well be considered representative on a national level, the ESTMJS recommendations are not fully equivalent on an international level. In contrast to the German guidelines, which are issued by the AWMF as umbrella organisation of the national scientific societies and are formally approved by the respective national boards of the interdisciplinary societies involved, the ESTMJS rather constitutes a more or less random sample of individuals. In addition, not all ESMTJS members participated actively in establishing the consensus. Even though the ESTMJS members are experts in the field of TMJ, mostly based at universities and/or scientifically active, they do not represent all European TMJ surgeons and do not hold a formal mandate to represent their national OMFS societies. As a consequence, the results of our study are less representative and robust and may not be generally applicable for other countries worldwide. They may, however, well constitute evidence of a trend. In a next step, therefore, this guideline should strive for approval by the respective national European societies.

Once more thank you very much for your thorough and helpful review. We hope we were able to implement your comments in an appropriate manner and to explain why in some cases, we choose not carry out some of the  suggested revisions and hope to meet with your approval according to the rebuttals given above.

Reviewer 2 Report

This article report an interesting scientific reflection on condylar dislocation's management leading to a series of recommandation. Its aim is to give a an international practice guideline based on the appreciation of a scientific society: the ESTMJS.

The recommendations listed here are, in my opinion, completely in accordance with current practices and with the literature on the subject; Furthermore, the absence of international recommendations on TMD management justifies the work carried out.

However some points should be revised.

The methodology of the initial guidelines draft is not very clear. It looks like they come from the German S3 interdisciplinary guidelines on Condylar Dislocation (lines 114-115) but then what is the point of the literature search. It is indeed expected to have an impact on this initial guidelines draft.

There are some uncertainties in the number of initial recommandations: 32 line 176 and 30 lines 180 and 184; whereas I understood that there were 30 recommandations at the initial voting and then 2 were added for the final one; This should be clarified

Adding sub-titles to clarify the sequence of evaluations (guidelines draft, preliminary vote then final vote) should help to clarify the assessment method

Discussion should focus on discussion! Indeed, a big part of its content consists in rephrasing the results. Appart from the nomenclature that is well explained, what readers should like to know specifically is:

-why this double voting methodology

-in which point these recommandations change from the old ones,

-and what the recent literature has added in TMD management

All this is more or less explained in the discussion but distilled into a repetition of the recommandations. 

Finally, abbreviations should be revised, we feel a bit lost between GoR, GoE, LoE, SoC, SoR, etc. And the head line of table 4) should be modified (SoC1 in the 5th column should be SoC2)

Author Response

Thank you very much for your kind recommendations.

Query: The methodology of the initial guidelines draft is not very clear. It looks like they come from the German S3 interdisciplinary guidelines on Condylar Dislocation (lines 114-115) but then what is the point of the literature search. It is indeed expected to have an impact on this initial guidelines draft.

Answer: We have amended our method section to clarify the development of the draft version of our recommendations. Please also cf. answers to reviewer #1. It applies that the literature search from 2014-2016 was conducted in the course of establishing the German guideline on condylar dislocation, which had been voted and agreed on in 2016 by a German interdisciplinary task force holding mandates from the respective scientific societies involved. This guideline then was translated into English to form the draft version the ESTMJS members voted on in 2019. This draft version represented a discussion base to be modified and discussed by the ESTMJS members in preparation of the General Assembly (GA); to do so, the draft version underwent a Delphi process among the ESTMJS members to evaluate (a) the extent of consensus among the ESTMJS members, (b) to identify controversial topics and then (c) a consensus was reached during a moderated consensus meeting held on the occasion of the GA. In preparation of the consensus meeting, to establish the ESTMJS guidelines a systematic literature update was performed in 2019 (by LS, MR and AN) and thus an updated  version of the draft was sent to all participating surgeons to keep the guideline up to date. In addition, any new literature with relevance for the recommendations was discussed during the consensus meeting. (e.g. supine vs sitting position, surgical procedures etc.) Finally, before finalizing the paper, another systematic  literature update was performed (LS, MR and AN)  and before submission to JCM, the recommendations established in 2019 were once more  approved by the president of the ESTMJS (F.S.) and then by all co-authors before submission to be still up to date in view of the literature published since the consensus meeting in 2019.

We modified (lines 106-126, in red):

  1. Materials and Methods

The consensus was formed using a modified Delphi methodology [6], following the principles of the Arbeitsgemeinschaft der Wissenschaftlichen Medizinischen Fachgesellschaften (AWMF, i.e. study group of the German scientific medical societies) [7] as follows;

2.1. Guideline draft

An initial set of draft guidelines were formulated, from the German S3 interdisciplinary guidelines on Condylar Dislocation (AWMF registry 007-063, June 2016 [1;5]), translated into English. This German S3 (i.e., evidence and consensus-based) guideline (first author and guideline coordinator A.N. [5]) is based on a systematic literature search, using the term “temporomandibular joint dislocation” in PubMed, Cochrane, Embase, and ZB MED databases which was originally conducted in 2014, 2015 and 2016. The literature research methodology according to the AWMF rules for S3 guidelines [7] is basically congruent to the PRISMA checklists and the guidelines are structured using PICOTS (for further details cf. long version and guideline report of the German guideline on Condylar dislocation, AWMF registry 007-063 under https://www.awmf.org/leitlinien/detail/ll/007-063.html) [5]). Two independent authors (U.V. and L.S.) screened all papers. A third author (A.N.) was consulted in cases of disagreement between these two independent screening results. Papers identified were graded by their level of evidence based on the criteria of the Oxford Centre for Evidence-based Medicine [8] and according to the rules of the AWMF [7]. The literature search was repeated in 2019 (M.R., L.S. and A.N.), to assess any new publications that might affect the recommendations, and again in 2020 (M.R., L.S. and A.N.) and 2021 (M.R. and A.N.) in preparation of this manuscript.

2.2. Delphi procedure (preliminary voting)

The initial set of draft guidelines consisted of 30 individual recommendations [5] relating to the assessment and treatment of TMJ dislocation. The expert group, consisting of members of the ESTMJS (cf. www.estmjs.org and eTable 1), were sent this initial draft 6 weeks prior to the general assembly (GA) held in Marburg, Germany, in May 2019. They were invited to grade these and make suggestions) for alterations and modifications (i.e., blinded to the other participants), which were processed by an independent monitor (L.S.).

We also added a section header for the 2.3. final voting (consensus meeting), to make the procedure more transparent.

Query: There are some uncertainties in the number of initial recommandations: 32 line 176 and 30 lines 180 and 184; whereas I understood that there were 30 recommandations at the initial voting and then 2 were added for the final one; This should be clarified

Answer: Thank you very much for mentioning the inconsistent count of recommendations (30 – 32), this was indeed a mistake we hadn’t noticed and for which we apologize.

Query: Adding sub-titles to clarify the sequence of evaluations (guidelines draft, preliminary vote then final vote) should help to clarify the assessment method

Answer: Thank you very much for this recommendation, we added subtitles to clarify the sequence of evaluations (please see above).

Query: Discussion should focus on discussion! Indeed, a big part of its content consists in rephrasing the results.

We are aware that our discussion is indeed quite long – still we think it is  crucial to explain to the readers, why certain recommendations were changed in what way and also to point out the major controversies and most relevant recommendations to the clinicians. We do hope you can agree.

Query: Apart from the nomenclature that is well explained, what readers should like to know specifically is:

-why this double voting methodology

You mention a „double voting methodology“ - does this criticism aim at the blinded voting via DELPHI procedure in the first place and then the final vote in the moderated consensus session? The purpose of these rules given by  the AWMF (i.e. study group of the German scientific medical societies) is explained in the last part of the discussion section.

Quotation: The blinded preliminary vote methodology of the modified DELPHI-procedure allows for anonymous comments and remarks and heightens the individual’s opinion, and therefore serves to collect all themes worth discussing, and identifying initial trends. It can however result in significant heterogeneity in voting, as demonstrated here where there was significant discordance within the preliminary voting, especially when compared to the homogeneity achieved with the German S3-guideline which served as a draft document. […] At the general assembly of the ESTMJS, participants were conversely exposed to interpersonal social and situational factors of influence including the non-blinded immediate result of the poll, resulting in potential bias by peer group pressure. The moderated consensus session offered the opportunity to discuss, reassess and change various recommendations or text passages, leading to a draft which could be agreed on unanimously. An independent moderator was used in our study according to the rules of the AWMF [7] to prevent over-emphasis of certain opinions in the discussion.

Query:

-in which point these recommendations change from the old ones,

Answer: All changes made to the draft version (which consists of the recommendations of the German guidelines, labelled there as the „old version“ of recommendations) were presented in Table 4, which was now moved to supplementary material as eTable 2 according to the  request of reviewer #1 (cf. above)

Query: -and what the recent literature has added in TMD management

All this is more or less explained in the discussion but distilled into a repetition of the recommendations.

Answer: Please cf. rebuttal to reviewer #1 above: as pointed out above, in 2019 all recommendations were up to date to the then existing literature. To keep our literature up to date, another literature search was performed in 2020 (LS, MR and AN) and a final update in 2021 (MR and AN) before submission took place and the recommendations established in 2019 were once more checked for plausibility and approved by all of our co-authors before submission.  

We corrected figure 2 (now including the latest literature update 2021, please take our apologies for missing to update the figure before submission)

Considering more recent developments reported in literature these did in fact not alter our recommendations and we discussed - if applying - why the ESTMJS members didn’t follow the conclusions given by some of the recent publications: Basically, what was more recently discussed with regard to condylar dislocation is summarized below:

  1. Autologous blood injection: this by now well evidence based procedure was already well considered during the consensus procedure: there were some new studies with high LoE such as Aamir et al. 2020, Ib-; Abrahamsson et al. 2019, Ia+; Bukhari und Rahim 2020, IIb+, all supporting the previous works by Daif 2010, Ib+; Hegab 2013, Ib+; Machon et al. 2018, Ib++; Oshiro et al. 2014, IIb+). The same applies for injections with botulinum toxin (Yoshida 2018b, IIb+)., so no changes with regard to our recommendations both for autologous blood injections and botulinum toxin, both representing safe and efficient treatment options for recurrent condylar dislocation.
  2. Recently, Jeyaraj et al. und Ihab et al. each published an RCT regarding surgical treatment options for recurrent dislocation (Ihab et al. 2020a, Ib+; Jeyaraj 2018, Ib+). Jeyaraj et al. (RCT with 75 patients, 25 patients per group,) compared eminectomy, eminectomy in combination with discoplasty and Dautrey’s procedure and overall showed more stable  and satisfying results for the Dautrey procedure. Nevertheless the authors considered eminectomy to be a safe and efficient option, as well, especially in combination with discoplasty (Jeyaraj 2018, Ib+). In conclusion, eminectomy was considered to be less invasive and therefore can be recommended. This is also confirmed by Ihab et al. (RCT with just 10 patients, bilaterally recurrent dislocation) treated with different augmentative procedures compared with eminectomy (Ihab et al. 2020a, Ib+). As a consequence, the ESTMJS members - though favouring eminectomy – abstained from recommending a certain type of surgery. Overall, the surgical recommendations still need more evidence based data to support any of the procedures (cf. Abrahamsson et al. 2019, Ia+; Tocaciu et al. 2019, V/k++).

Query: Finally, abbreviations should be revised, we feel a bit lost between GoR, GoE, LoE, SoC, SoR, etc. And the head line of table 4) should be modified (SoC1 in the 5th column should be SoC2)

Answer: Thank you for the correction of the header, we corrected the mistake accordingly (now eTable 3).
We also understand that our frequent-used abbreviations can cause confusion – additionally to the explanation in the text, we therefore amended a supplemented list of abbreviations (eTable 2).

Thank you very much for your thorough and helpful review. We hope we were able to implement your comments in an appropriate manner and to explain why in some cases, we did not carry out the suggested revisions and hope to meet with your approval according to the rebuttals given above.

Reviewer 3 Report

Very high quality, well-designed study by the ESTMJS.  The authors are to be commended for the effort and longitudinal follow-up for the society's consensus guidelines to be formed.  This will provide a great service to the field to have this level of evidence added to the TMJ literature.  

Author Response

Thank you very much for your praise and encouraging words. We are very pleased if we are given the chance to  publish this work and think it will be useful for our readers.